

# Gametophytic self-incompatibility in Andean capuli (*Prunus serotina* subsp. *capuli*): allelic diversity at the S-RNase locus influences normal pollen-tube formation during fertilization

Milton Gordillo-Romero[*], Lisa Correa-Baus[*], Verónica Baquero-Méndez, María de Lourdes Torres, Carlos Vintimilla, Jose Tobar and Andrés F. Torres

Laboratorio de Biotecnología Vegetal, Colegio de Ciencias Biológicas y Ambientales, Universidad San Francisco de Quito, Quito, Pichincha, Ecuador

[*] These authors contributed equally to this work.

Corresponding author
Andrés F. Torres, atorres@usfq.edu.ec

## ABSTRACT

Capuli (*Prunus serotina* subsp. *capuli*) is a tree species that is widely distributed in the northern Andes. In *Prunus*, fruit set and productivity appears to be limited by gametophytic self-incompatibility (GSI) which is controlled by the S-Locus. For the first time, this research reveals the molecular structure of the capuli S-RNase (a proxy for S-Locus diversity) and documents how S-Locus diversity influences GSI in the species. To this end, the capuli S-RNase gene was amplified and sequenced in order to design a CAPS (Cleaved Amplified Polymorphic Sequence) marker system that could unequivocally detect S-alleles by targeting the highly polymorphic C2–C3 S-RNase intra-genic region. The devised system proved highly effective. When used to assess S-Locus diversity in 15 *P. serotina* accessions, it could identify 18 S-alleles; 7 more than when using standard methodologies for the identification of S-alleles in *Prunus* species. CAPS marker information was subsequently used to formulate experimental crosses between compatible and incompatible individuals (as defined by their S-allelic identity). Crosses between heterozygote individuals with contrasting S-alleles resulted in normal pollen tube formation and growth. In crosses between individuals with exactly similar S-allele identities, pollen tubes often showed morphological alterations and arrested development, but for some (suspected) incompatible crosses, pollen tubes could reach the ovary. The latter indicates the possibility of a genotype-specific breakdown of GSI in the species. Overall, this supports the notion that S-Locus diversity influences the reproductive patterns of Andean capuli and that it should be considered in the design of orchards and the production of basic propagation materials.

## INTRODUCTION

The capuli (*Prunus serotina* subsp. *capuli*) is an allotetraploid woody perennial species, taxonomically positioned in a clade with the *Padus* and *Laurocerasus* sub-genera of the

*Prunus* genus (*Pairon, Potter & Jacquemart, 2008*; *Bortiri, Vanden & Potter, 2006*; *Guzmán et al., 2018*). The species is native to North America, but has been naturalized in the Andean highlands of Colombia, Ecuador, Peru and Bolivia (*Fresnedo-Ramírez, Segura & Muratalla-Lúa, 2011*; *Guzmán et al., 2018*). *P. serotina* was likely introduced in the 17th century during the Spanish colonization of South America and has since developed a rich ethnobotanical tradition in the region (*Popenoe & Pachano, 1922*). Today, local communities of the Andes cultivate *P. serotina* for its sweet black drupes. These are used to produce traditional delicacies, ceremonial drinks and home remedies (*Popenoe & Pachano, 1922*).

*P. serotina* has the potential to contribute to the development of profitable, resilient and biodiverse farming systems in the high Andes. In addition to its palatable flavor, the capuli drupe possesses antimicrobial (*Jiménez et al., 2011*), antioxydant (*Luna-Vázquez et al., 2013*) and anti-inflammatory properties (*Vasco et al., 2009*; *Álvarez Suárez et al., 2017*). These characteristics make it an attractive product for health-food and nutraceutical markets. The export potential of the capuli fruit could serve as a socio-economic incentive towards the conservation, potentiation and utilization of this semi-domesticated species, while simultaneously improving rural livelihoods (*National Research Council, 1989*).

The effective introduction of *P. serotina* in commercial orchards will require important interventions in genetics and agronomy. In particular, understanding the species' gametophytic self-incompatibility (GSI) system will prove crucial to the design of orchards that maximize fruit set and yields. GSI is a mechanism that has evolved to inhibit self-pollination and to restrict cross-pollination between genetically related individuals, thus preventing inbreeding depression while promoting heterozygosity (*Ushijima et al., 1998*; *De Nettancourt, 2001*). In *Prunus*, GSI is controlled by a single, multi-allelic locus (i.e., the S-Locus). The S-Locus is composed of two independent genes that are tightly linked: the style-specific ribonuclease (S-RNase) and the S-locus F-box protein (SFB). These genes respectively encode for the pistil and pollen specificity factors of the self-incompatibility response (*Tao & Iezzoni, 2010*). Individuals showing the same S-haplotype (i.e., identical alleles for the S-RNase and SFB genes) belong to the same incompatibility group (IG) and display cross-incompatibility. In numerous *Prunus* species, GSI poses a restriction to free pollen flow and must be considered in the selection of cross-compatible individuals to ensure fruit set in commercial orchards (*Herrera et al., 2018*). GSI can also influence the selection of compatibility groups in breeding programs and seed multiplication schemes (*Cachi et al., 2017*).

To our knowledge, GSI in *P. serotina* remains largely unexplored. *Donovan (1969)* studied pollen extension and fruit-set patterns in selfings and crosses of North American *P. serotina* populations. The results of this study provided support for GSI and some cross-incompatibility in the species, but did not exclude the possibility for genotype-specific self-compatibility. At the molecular level, *Gordillo et al. (2015)* investigated the allelic diversity of the S-Locus of Andean *P. serotina* by evaluating the degree of polymorphic variation in the Intron-I region of the S-RNase gene in 80 accessions from the Ecuadorian highlands. However, this study did not evaluate if and how this molecular diversity controls GSI in the species. Therefore, the main objective of this research was to study whether

the *P. serotina* S-Locus controls sexual incompatibility in the species. To this end, we characterized the molecular structure of the *P. serotina* S-RNase gene in order to design an efficient molecular marker system that would enable the analysis of the allelic diversity of the S-Locus in Andean capuli populations. Subsequently, S-allelic information was used to set-up experimental crosses between cross-compatible and cross-incompatible haplotypes in order to evaluate pollen tube formation patterns in-vivo. Ultimately, we aimed to demonstrate that *P. serotina* individuals from the same IG exhibit obstructed fertilization; the latter would support the notion that the S-Locus influences the reproductive patterns of the species.

## MATERIALS & METHODS

### Genetic and plant material

The characterization of the *P. serotina* S-RNase gene and the development of a CAPS (Cleaved Amplified Polymorphic Sequence) marker system were performed on a set of 15 *P. serotina* accessions for which S-RNase Intron-I polymorphisms had been previously described (*Gordillo et al., 2015*). Leaf samples for these accessions were originally collected by *Guadalupe et al. (2015)* from 6 provinces across the highlands of Ecuador. The geographical coordinates for the collection site of each accession are detailed in Table S1. DNA samples for all 15 accessions are stored at the Plant Biotechnology Laboratory of Universidad San Francisco de Quito (Cumbayá, Ecuador) and were used in this investigation.

For pollination assays, we used an independent set of 7 capuli trees located in a private orchard in Cayambe, Ecuador. These trees were selected because of their proximity to the Plant Biotechnology Laboratory of Universidad San Francisco de Quito. For each individual tree, genomic DNA was isolated from young leaves using the CTAB method (*Xin & Chen, 2006*). DNA concentration and quality were assessed by spectrophotometry using a Nanodrop 2000 spectrophotomer (Thermo Fisher Scientific).

### S-RNase gene amplification and sequencing

The amplification of the S-RNase gene of 15 *P. serotina* accessions previously studied by *Gordillo et al. (2015)* was performed using the PaConsI-F (MCTTGTTCTTGSTT-TYGCTTTCTTC; *Sonneveld, Tobutt & Robbins, 2003*) and EM-PC5consRD (CAAAAT-ACCACTTCATGTAACARC; *Sutherland, Robbins & Tobutt, 2004*) primers. These primers respectively target the Signal Peptide and C5 regions of the *Prunus* S-RNase gene. PCR reactions consisted of 1× PCR Buffer (Invitrogen), 0.2 mM dNTPs, 1.5 mM MgCl2, 0.3 μM of each primer, 1 U of Taq DNA polymerase (Invitrogen) and 40 ng of DNA in a 25 μl final volume. Amplification conditions were as described by *Ortega et al. (2006)*, but with a 58 °C annealing temperature. Amplified products were separated by gel electrophoresis in 1.5% agarose gels at 80 volts for 90 min using TBE 1× as running buffer. DNA fragments were visualized using SYBR-Safe (Invitrogen).

Finally, amplified products were excised from agarose gels and purified using the Wizard SV Gel and PCR Clean-Up System Protocol (Promega). Purified products were

subsequently reamplified by PCR using the amplification conditions described above. Reamplified products were sequenced at Macrogen Inc. (Seoul, Korea).

## Analysis of the *P. serotina* S-RNase sequences

Consensus nucleotide sequences from forward and reverse reads of all amplified fragments were obtained using PREGAP and GAP4 (*Staden, 1996*). The search for homologous sequences in the GenBank database was performed using the NCBI BLAST tool (*Altschul et al., 1990*). Intraspecific and interspecific alignments were performed using the ClustalW algorithm of MEGA7 (*Kumar, Stecher & Tamura, 2016*). Deduced amino acid sequences were obtained by translating the coding sequence corresponding to each of the alleles using the online ExPASy Tool (*Gasteiger et al., 2003*). Alignments of the deduced amino acid sequences were performed using the Clustal Omega algorithm of the European Bioinformatics Institute (*Madeira et al., 2019*). Ka/Ks ratios calculation and the sliding window analysis with a window length of 20 codons were performed using the DnaSP 6 Software (*Rozas & Rozas, 1999*). Shannon Entropy indexes were calculated for each of the conserved regions (C1, C2, C3, RC4) and the RHV, using the online Protein Variability Server (*García-Boronat et al., 2008*).

## Design and validation of the CAPS marker system

The CAPS marker system was designed *in-silico* using the Genome Compiler Software Package (*Amirav-Drory, Debbi & Nevo, 2015*). The digest function was employed for the screening of restriction sites located within the analyzed sequences using a pool of 100 available enzymes, and the gel simulation function was employed for the visualization of the restriction patterns in virtual gels after the *in-silico* digestions.

For the *in-vitro* validation of the developed CAPS marker system, we designed species specific primers to amplify the C2–C3 intergenic region of the *P. serotina* S-RNase gene: Ps1C2Fw (ATY CAT-GGC-CTR-TGG-CCA-AG) and Ps2C3Rv (TGY TTR-TTC-CAT-TCV-CBT-TCC). Primer design was based on S-RNase full-length sequences produced in this study. The C2–C3 regions of the 15 *P. serotina* accessions were amplified using the Ps1C2Fw/Ps2C3Rv primers. Reagents concentrations for the PCR reaction were the following: 1× PCR Buffer, 3 mM MgCl2, 0.2 mM dNTPs, 0.5 μM of each primer, 1 U of Taq polymerase Platinum (Invitrogen) and 20 ng of DNA in a 25 μl final volume. Cycling conditions consisted in: 2 min of initial denaturation at 94 °C, followed by 35 cycles of 1 min at 94 °C, 2 min at 58 °C, 4 min at 68 °C and 10 min of final extension at 68 °C. PCR products were separated by gel electrophoresis in 1.5% agarose gels. Running conditions were 80 volts for 90 min using TBE 1× as running buffer. DNA fragments were visualized using SYBR-Safe (Invitrogen). All the resulting bands were excised from the agarose gel and the DNA was recovered using the Wizard SV Gel and PCR Clean-Up System (Promega). Purified DNA was reamplified using the same amplification conditions as described before using 40 ng of DNA as template. Reamplified fragments were digested with RsaI, MboI, HinfI from the Anza line of Thermo Fisher Scientific. The digestion reactions consisted in individual digestions containing 9 μl of nuclease-free water, 2 μL of digestion Buffer 10×, 1 μL of enzyme and 8 μL (<0.2 μg) of the reamplified fragments. Restriction reactions were

incubated at 37 °C for 20 min. Restriction patterns were visualized by gel electrophoresis in 2% agarose gels. Running conditions were 80 volts for 1 h using TBE 1× as running buffer. DNA fragments were visualized using SYBR-Safe (Invitrogen).

## Pollination Assays: pistil staining and analysis by fluorescence microscopy

To evaluate whether S-Locus allelic variation influences self-incompatibility responses in *P. serotina*, we studied pollen tube formation and growth patterns in experimental crosses. Seven *P. serotina* trees at flowering stage were S-haplotyped using the CAPS marker system developed in this study. Based on S-haplotype results, we conducted incompatible crosses (i.e., self-pollinations or crosses between 2 heterozygote individuals carrying the exact same S-alleles) and compatible crosses (i.e., between heterozygotes presenting completely different S-alleles) by hand pollination in the laboratory. To this end, branches with flowers at stage 59 of the BBCH scale (*Ramírez & Davenport, 2016*) were cut and transported to the Plant Biotechnology Laboratory of Universidad San Francisco de Quito and placed in water with plant food. To extract pollen from donor flowers, anthers were manually removed and dried for 24 h at room temperature to facilitate adequate pollen release. Collected pollen was kept at 4 °C until further use. For pollination experiments, receptor flowers were emasculated and pollinated using a paint brush 24 h after emasculation. Per experimental cross, 25 pistils were collected 48 h following hand pollination.

Subsequently, pollinated pistils were stained using the aniline blue protocol described by *Jefferies & Belcher (1974)*, and pollen tube formation and extension patterns were analyzed via fluorescence microscopy. For the aniline blue staining protocol, pollinated pistils were fixed in 70% ethanol for 7 days. As preparation for microscopy, pistils were incubated at 62 °C in 10 M NaOH for 40 min to soften the tissue and to enhance staining. Softened pistils were washed in distilled water and stained with aniline blue for 1 h. Pollen tube extension patterns were observed using a BX50 Fluorescent Microscope (Olympus). The Multiple Image Analysis software package (Amscope) was used to obtain images of the style and for counting pollen tubes. A total of 78 pistils were analyzed. All protocols were previously standardized specifically for *P. serotina* subsp. *capuli*.

## Data analysis of pollen tubes growth assay

In the style, pollen tube growth was scored as the number of pollen tubes that reached the lower-third section of the style. To compare pollen tube growth and development in the different type of crosses, the data was subjected to a normality test and to a Levene's test for assessing the homocedasticity of the data. Subsequently, a one-way ANOVA was carried out and the means were separated using the Fisher Pairwise comparisons, grouping information using the Fisher LSD Method and 95% of confidence. Analysis were performed using *Minitab* 17 Software package *(2010)*.

# RESULTS

## Molecular structure of the *P. serotina* S-RNase gene

In this study, we amplified, sequenced and characterized the S-RNase gene of 15 distinct *P. serotina* accessions. The amplification process generated a total of 22 DNA

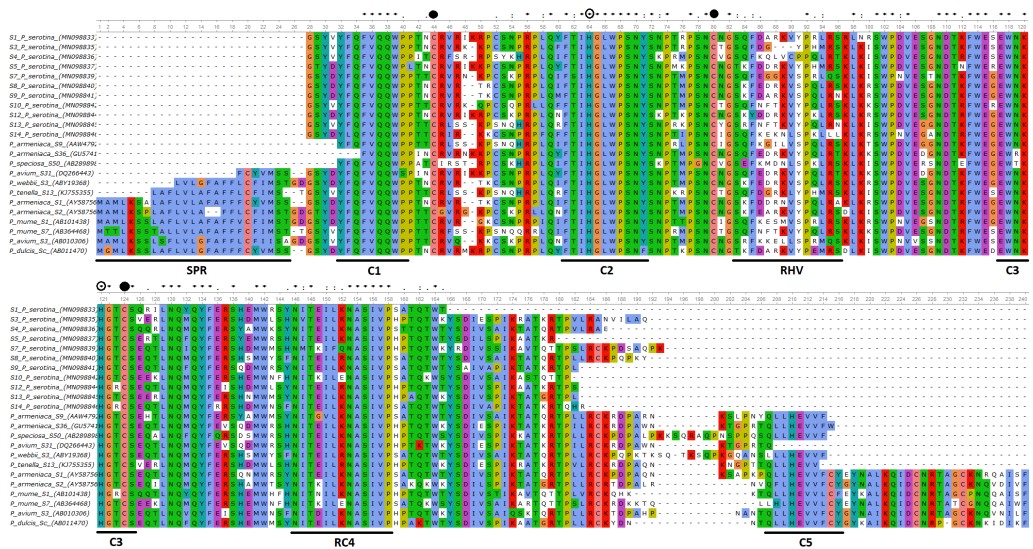

**Figure 1** **Alignment of the deduced amino acid sequences of *P. serotina* S-RNases with S-RNases reported for different *Prunus* species.** Alignment included: *P. serotina* (capuli), *P. tenella* (dwarf Russian almond), *P. armeniaca* (apricot), *P. salicina* (plum), *P. mume* (Japanese apricot), *P. avium* (sweet cherry) for determining the genetic structure of the *P. serotina* S-RNase gene. Aligning algorithm: Clustal Omega (EMLB-EBI). Asterisks denote conserved sites; dots denote conservative substitutions and dashes indicate gaps. Conserved cysteine residues are pointed out with a filled circle, whereas conserved histidine residues are pointed out with an open circle. The signal peptide, the conserved (C1–C5) and the hyper-variable (RHV) regions are underlined. GenBank accession numbers: *P. ten* $S_{13}$ (KJ755355), *P. sal* $S_c$ (AB084102), *P. avi* $S_{31}$ (DQ266443), *P. arm* $S_{36}$ (GU574198), *P. arm* $S_1$ (AY587561), *P. arm* $S_2$ (AY587562), *P. mum* $S_1$ (AB101438), *P. mum* $S_7$ (AB364468), *P. avi* $S_3$ (AB010306), *P. sal* $S_a$ (AB252411), *P. dul* $S_c$ (AB011470).

fragments ranging in size from 1023 to 2144 base pairs (Fig. S1). The identity of these 22 amplified fragments was evaluated by comparing their Intron-I sequence with Intron-I sequences reported by *Gordillo et al. (2015)*. These comparisons revealed that 19 of the 22 amplicons corresponded to the 11 alleles previously described by *Gordillo et al. (2015)*. The remaining 3 sequences were identified as new S-RNase alleles. We adopted standard nomenclature rules to catalogue the 14 detected alleles as *P. serotina* $S_1$–*P. serotina* $S_{14}$. DNA sequences for all identified alleles were submitted to the NCBI GenBank database with the following accession numbers: MN098833, MN098834, MN098835, MN098836, MN098837, MN098838, MN098839, MN098840, MN098841, MN098842, MN098843, MN098844, MN098845 and MN098846 (Table S2).

Deduced amino acid sequences for 11 of the 14 identified *P. serotina* S-RNase alleles identified were aligned and compared amongst them (intraspecific comparisons), as well as with S-RNase sequences retrieved for other *Prunus* species (interspecific comparisons) (Fig. 1). Alleles $S_2$, $S_6$ and $S_{11}$ were not included in these analyses given that only a small fraction of their coding sequences could be retrieved. Overall, intraspecific homologies ranged from ∼70% to ∼90%, while interspecific homologies ranged from ∼81% to ∼99%.

Furthermore, we confirmed that the *P. serotina* S-RNase gene exhibits the characteristic molecular structure of *Prunus* T2/S-type RNases. When comparing the deduced amino

acid sequences of *P. serotina* S-RNases with S-RNases reported for other *Prunus* species, we were able to identify the signal peptide region, Introns I and II, the five conserved regions and the hypervariable region located between C2 and C3 (Fig. 1). Shannon entropy indices ($H$) were employed to determine the degree of sequence variability for each of the gene's structural regions amongst the 11 evaluated alleles. Our results indicated that the least variable regions were C1, C2 and C3 ($H = 0.05$, $0.049$ and $0.09$, respectively), while the RC4 region presented a higher variability index ($H = 0.28$). The C5 region was not included in the analysis as it corresponded to the primer anchoring site (used for sequencing); therefore, its sequence could not be retrieved. As anticipated, the most variable region was the C2–C3 intragenic region ($H = 1.28$), containing the RHV region. Our results also indicated that the RHV sequence was unique for each of the 11 examined S-RNase alleles (Fig. 1). In addition, Ka/Ks ratios were used to identify regions with a higher likelihood for the occurrence of non-synonymous mutations along the S-RNase coding sequence. A sliding-window analysis of Ka/Ks ratios for the 11 examined alleles showed that the highest Ka/Ks ratio (1.34) was located within the RHV region. Conversely, the C2 and C3 regions showed the lowest Ka/Ks ratios (0.37 and 0.17, respectively) (Fig. 2).

## Development of a CAPS marker system for analyzing the allelic diversity of the S-Locus in *P. serotina*

Given its high degree of polymorphism in size and sequence, the C2–C3 intragenic region was used as a target for the development of a CAPS (Cleaved Amplified Polymorphic Sequence) marker system for the identification of S-RNase alleles in *P. serotina*. The CAPS marker system entails the amplification of a specific DNA region followed by enzymatic restriction to generate unique and distinctive restriction patterns (*Shavrukov, 2016*).

The design of the CAPS marker system was performed *in-silico* using the Genome Compiler Software Package. Sequences corresponding to the C2–C3 regions (isolated from full-length DNA sequences of the 11 S-RNase alleles) were screened using 100 restriction enzymes. From these, we identified 9 restriction enzymes with the ability to cleave all or the majority (>~80%) of the 11 sequences analyzed. Based on these results, 99 *in-silico* restrictions (11 alleles × 9 enzymes) were simulated and the digestion patterns were visualized using the virtual simulator of Genome Compiler. These analyses demonstrated that to unequivocally identify the 11 S-alleles, at least 3 restriction enzymes were necessary (RsaI, MboI, HinfI). Each of these enzymes generated individual digestion patterns (Table 1) that when read simultaneously, produced a unique fingerprint for each S-allele. Based on this selection of enzymes and the digestion patterns, we generated a CAPS guide for the rapid identification of *P. serotina* S-RNase alleles (Table 2).

*In-silico* results were then validated in-vitro by amplifying the C2–C3 region corresponding to the 15 *P. serotina* accessions, followed by independent enzymatic restrictions with RsaI, MboI and HinfI. While the expectation was to obtain a maximum of 14 alleles for these 15 accessions (based on DNA sequence analysis), 4 additional (unexpected) bands were obtained for accessions H25, Azu15, Pic19, Car7, Car5, Car3, and Car12 during the C2–C3 amplification (Fig. 3). After the enzymatic digestion, the restriction patterns of the already characterized alleles matched with the *in-silico* predictions. Digestion

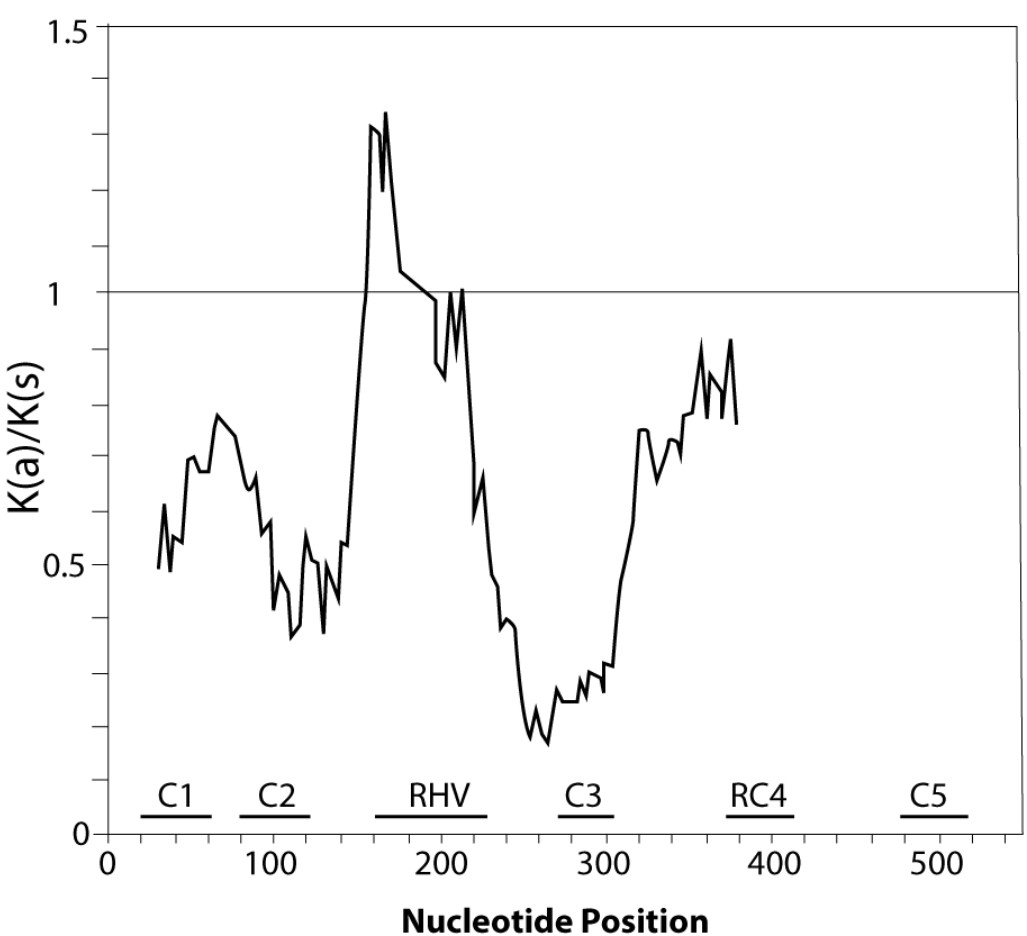

**Figure 2** Graphical representation of the Ka/Ks ratios calculated along the coding sequence of the 11 *P. serotina* S-alleles based on a sliding window analysis of 20 codons. The position of conserved regions (C1–C5) and RHV region are indicated. The plot was obtained using DnaSP6 (*Rozas & Rozas, 1999*).

patterns of the 4 unexpected bands did not match with any of our *in-silico* predictions, therefore we presumed that these bands corresponded to new uncharacterized S-alleles, which were designated as $S_{15}$–$S_{18}$. CAPS restriction patterns corresponding to the $S_1$–$S_{18}$ alleles are presented in Fig. 4.

## Influence of S-locus allelic variation on pollen tube formation patterns

To evaluate whether S-Locus allelic variation influences self-incompatibility responses in *P. serotina*, we studied pollen tube formation and growth patterns in pollination experiments. To this end, we selected 7 *P. serotina* tress from a private orchard in Cayambe (Ecuador) based on their S-allelic composition (as revealed with our CAPS marker system). The 7 individuals, which were in full bloom and exhibited flowers at stage 59 of the BBHC scale, harbored 9 distinct S-alleles. Six of these alleles had been previously characterized ($S_1$, $S_4$, $S_6$, $S_8$, $S_9$, $S_{10}$), but the remaining 3 patterns corresponded to new uncharacterized alleles; the latter were designated as $S_{19}$–$S_{21}$ (Table 3).

Table 1 Individual digestion patterns expected for the digestion of C2–C3 amplicons with three different enzymes (RsaI, MboI and HinfI). Composed patterns (formed by the simultaneous reading of the three individual digestion patterns) specific to each S-allele are presented in Table 2.

| Pattern | Restriction enzymes | | | | | | | | | | | |
|---|---|---|---|---|---|---|---|---|---|---|---|---|
| | RsaI | | | | MboI | | | | HinfI | | | |
| A | 200 | | | | 200 | | | | 180 | 100 | | |
| B | 200 | 160 | 150 | | 210 | 200 | 190 | | 200 | | | |
| C | 210 | 140 | 100 | | 240 | | | | 225 | 200 | | |
| D | 210 | 180 | 160 | | 300 | | | | 230 | 190 | 100 | |
| E | 230 | 200 | | | 300 | 100 | | | 240 | | | |
| F | 240 | | | | 360 | | | | 250 | 210 | | |
| G | 240 | 225 | | | 460 | | | | 360 | 250 | 220 | |
| H | 290 | 170 | 100 | | 465 | 250 | 230 | | 365 | | | |
| I | 330 | 210 | 180 | | 490 | | | | 375 | 330 | 250 | 210 |
| J | 370 | | | | 520 | 430 | | | 380 | | | |
| K | 430 | 250 | | | 550 | 415 | 250 | | 390 | 240 | | |
| L | 475 | 360 | 290 | | 600 | 470 | 300 | | 470 | | | |
| M | 620 | 210 | 195 | 170 | 615 | | | | 540 | | | |
| N | 790 | 510 | | | 760 | | | | 550 | 495 | | |
| O | 940 | 220 | 170 | | 1,200 | | | | 550 | 350 | | |
| P | 950 | | | | 290 | 180 | 100 | | 580 | | | |
| Q | 280 | 120 | 90 | | 600 | 500 | 400 | 380 | 650 | 330 | 270 | |
| R | 220 | 200 | 110 | | 430 | 100 | | | 290 | 220 | | |

Notes.
In order to facilitate the digestion pattern identification, fragment sizes presented in this table were rounded.

Based on these results, 2 types of crosses were established: incompatible crosses (i.e., self-pollinations or crosses between 2 heterozygote individuals carrying the exact same S-alleles) and compatible crosses (i.e., between heterozygote individuals presenting completely different S-alleles). Crossing assays are summarized in Table 4. To evaluate incompatible and compatible responses, we analyzed pollen tube extension patterns via fluorescence microscopy. These results clearly indicated that pollen tubes develop differently depending on the type of cross (Fig. 5). In compatible crosses, we found pollen tubes that reached the ovary in all pollinated pistils (Fig. 5A). By contrast, in the pistils of incompatible crosses, pollen tube growth was typically inhibited in the upper- and middle-third sections of the style (Fig. 5B). Furthermore, pollen tubes in incompatible crosses showed swelling at the tip (Fig. 5C). This morphological alteration was not observed in compatible crosses. For specific incompatible crosses, several pollinated pistils displayed a variable number of pollen tubes that reached the bottom-third section of the style, but in consistently lower proportions relative to compatible crosses (Table S3). The occurrence of this phenomenon was especially common in crosses 22 × 17 and 17 × 17. It is important to highlight, nevertheless, that pollen tubes that did reach the lower-third segment of the style in incompatible crosses always showed swollen tips. A one-way analysis of variance confirmed that differences in pollen tube growth (based on arrested development vs. full development to reach the ovary) between incompatible and compatible crosses were statistically significant $p < 0.001$.

**Table 2  Composed digestion patterns for the identification of P. serotina S-alleles ($S_1$–$S_{18}$) using three restriction enzymes.** Each letter refers to the restriction pattern shown in Table 1 for each restriction enzyme (RsaI, MboI and HinfI, respectively).

| S-RNase allele | Restriction pattern |
|---|---|
| $S_1$ | PJN |
| $S_3$ | BEJ |
| $S_4$ | OLI |
| $S_5$ | DFC |
| $S_6$ | CIE |
| $S_7$ | EGL |
| $S_8$ | LHQ |
| $S_9$ | IMK |
| $S_{10}$ | KNJ |
| $S_{12}$ | JFF |
| $S_{13}$ | HCP |
| $S_{14}$ | GBM |
| $S_{15}$ | FCA |
| $S_{16}$ | MKG |
| $S_{17}$ | FCE |
| $S_{18}$ | NOO |

**Table 3  S-allelic composition of the 7 *P. serotina* individuals selected for performing assisted crosses.** All individuals were heterozygotes and amplified 2 bands. S-alleles were determined using the CAPS marker system.

| Individual | C2–C3 CAPS Patterns | | | | S-allele |
|---|---|---|---|---|---|
| | Band | RsaI | MboI | HinfI | |
| 1 | 1 | L | H | Q | $S_8$ |
| | 2 | Q | P | R | $S_{19}$ |
| 12 | 1 | O | L | I | $S_4$ |
| | 2 | I | M | K | $S_9$ |
| 13 | 1 | O | L | I | $S_4$ |
| | 2 | I | M | K | $S_9$ |
| 14 | 1 | O | Q | I | $S_{20}$ |
| | 2 | R | R | F | $S_{21}$ |
| 15 | 1 | K | N | J | $S_{10}$ |
| | 2 | C | I | E | $S_6$ |
| 17 | 1 | O | L | I | $S_4$ |
| | 2 | P | J | N | $S_1$ |
| 22 | 1 | O | L | I | $S_4$ |
| | 2 | P | J | N | $S_1$ |
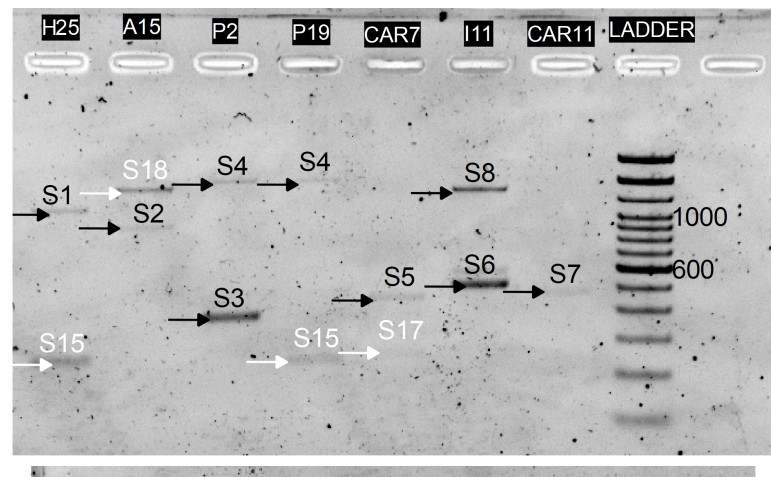

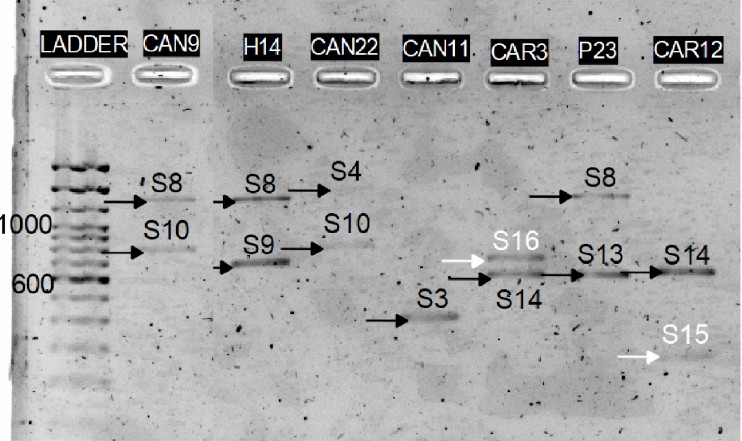

**Figure 3** **PCR amplification of the *P. serotina* C2–C3 intragenic region using the Ps1C2Fw and Ps2C3Rv primers.** Black arrows indicate the position of the expected alleles whereas white arrows highlight the unexpected alleles obtained for H25, Azu15, Pic19, Car7, Car3 and Car12.

**Table 4** **Proposed crosses according to the S-allelic composition of the 7 *P. serotina* trees genotyped.** Crosses between two individuals carrying the exact same S-alleles are expected to be incompatible. Crosses between two individuals carrying completely different S-alleles are expected to be compatible. Self-pollinations of the pollen receptor trees are expected to be incompatible.

| Pollen donor | | Pollen receptor | Expected phenotype | Observed phenotype |
|---|---|---|---|---|
| 22 ($S_1$, $S_4$) | × | 17 ($S_1$, $S_4$) | Incompatible | Undetermined |
| 12 ($S_4$, $S_9$) | × | 13 ($S_4$, $S_9$) | Incompatible | Incompatible |
| 17 ($S_1$, $S_4$) | × | 1 ($S_8$, $S_{19}$) | Compatible | Compatible |
| 14 ($S_{20}$, $S_{21}$) | × | 15 ($S_{10}$, $S_6$) | Compatible | Compatible |
| 17 ($S_1$, $S_4$) | × | 17 ($S_1$, $S_4$) | Incompatible | Undetermined |
| 13 ($S_4$, $S_9$) | × | 13 ($S_4$, $S_9$) | Incompatible | Incompatible |
| 1 ($S_8$, $S_{19}$) | × | 1 ($S_8$, $S_{19}$) | Incompatible | Incompatible |
| 15 ($S_{10}$, $S_6$) | × | 15 ($S_{10}$, $S_6$) | Incompatible | Incompatible |
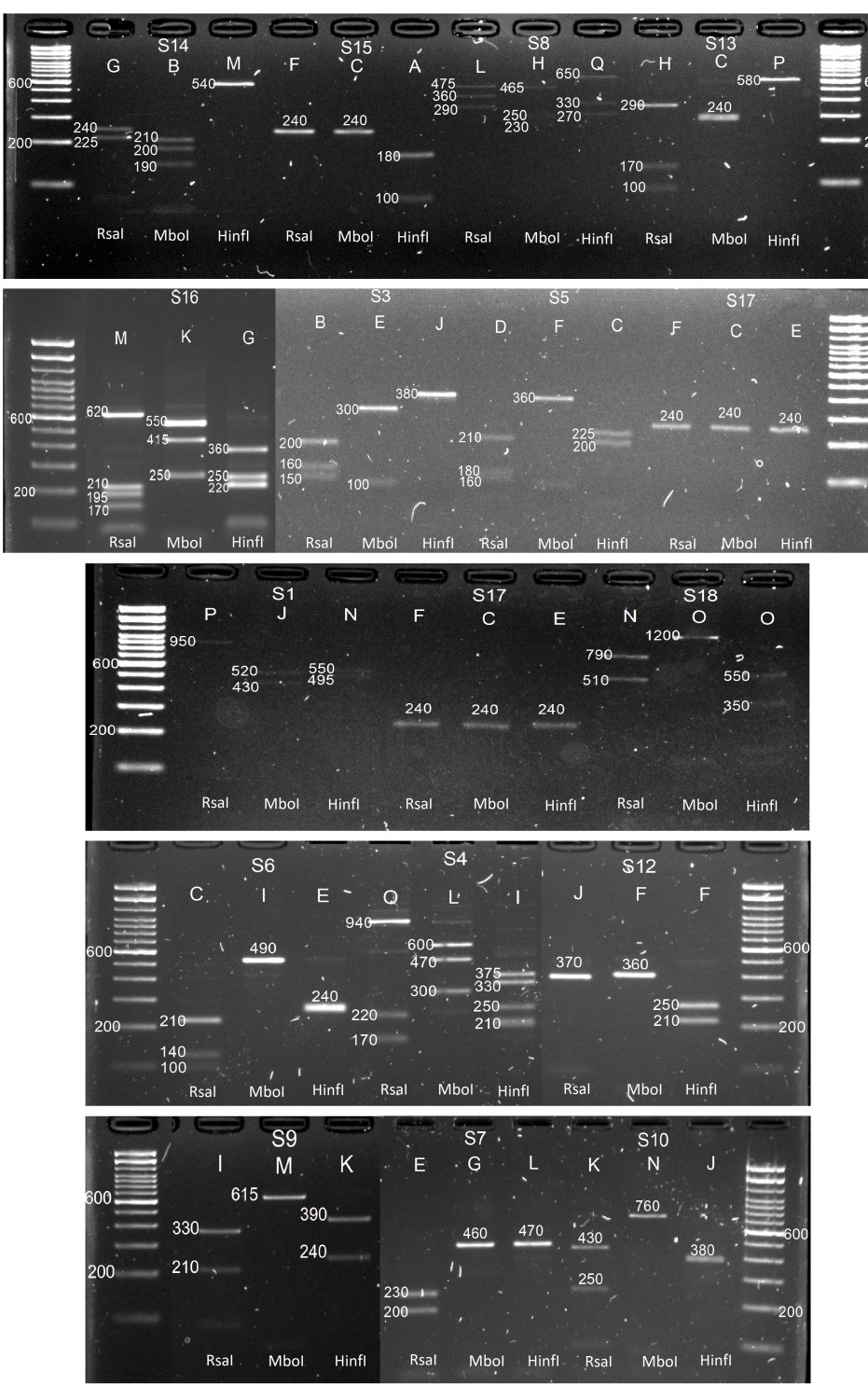

**Figure 4  In-vitro CAPS patterns obtained for alleles S₁, S₃, S₄, S₅, S₆, S₇, S₈, S₉, S₁₀, S₁₂, S₁₃, S₁₄, S₁₅, S₁₆, S₁₇ and S₁₈.** Letters indicate the restriction patterns reported for each enzyme in Table 1: RsaI, MboI and HinfI, respectively.

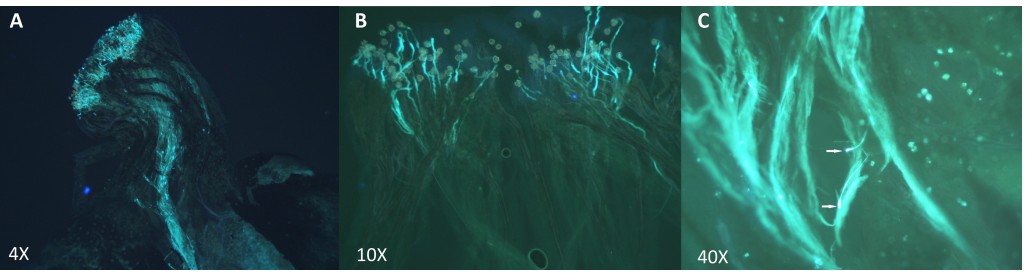

**Figure 5** **Pollen tube growth analysis in incompatible and compatible assisted crosses using fluorescence microscopy.** Pollen tubes were stained following the aniline blue protocol after 48 h of hand pollination. (A) Compatible cross: individual 17 ($S_1$–$S_4$) × Individual 1 ($S_8$–$S_{19}$). Pollen tubes grew along the style and reached the ovary. (B) Incompatible cross: Self-pollination of Individual 1 ($S_8$–$S_{19}$). Pollen tube growth was inhibited in the upper-third section of the style. (C) Swollen tips of Self-pollination of Individual 13 ($S_4$–$S_9$).

## DISCUSSION

### The S-RNase gene in *P. serotina* resembles those of related species

The main objective of this research was to elucidate whether the S-Locus of *P. serotina* controls sexual incompatibility patterns in the species. To this end, we used the S-RNase gene as a proxy for the identification of S-haplotypes in *P. serotina* and evaluated whether S-RNase allelic diversity correlates with sexual incompatibility patterns.

Detailed analysis of the amino acid sequences of 11 *P. serotina* S-RNase alleles confirms that the *P. serotina* S-RNase gene has a high degree of amino-acid sequence homology (∼81% to ∼99%) with stylar ribonucleases from related *Prunus* species (*Ushijima et al., 1998*; *Yaegaki et al., 2001*; *Vaughan et al., 2008*; *Gao et al., 2012*). For instance, the *P. serotina* $S_8$ and *P. webbii* $S_3$ (NCBI: ABY19368) alleles displayed a protein identity of ∼99% (Fig. S2). A high degree of interspecific amino acid sequence homology between S-RNase alleles in *Prunus* has been reported earlier (*Banovic et al., 2009*; *Ortega et al., 2006*; *Surbanovski et al., 2007*). These findings support the hypothesis that *Prunus* S-alleles with very high interspecific identities (>99%) may have originated from a common ancestor before speciation, and that these alleles have been conserved ever since in the derived lineages (*Ortega et al., 2006*; *Surbanovski et al., 2007*; *Sassa et al., 1996*).

### CAPS as a molecular marker for determining the allelic diversity of the S-Locus

To investigate GSI-controlled incompatibility patterns in *P. serotina*, we required an effective marker system that would enable the unequivocal identification of S-RNase allelic diversity. The standard methodology for S-alleles identification in *Prunus* species involves the amplification of the S-RNase Intron I and Intron II, followed by amplicon size differentiation via gel electrophoresis (*Ortega et al., 2006*; *Sonneveld, Tobutt & Robbins, 2003*; *Gao et al., 2012*). However, this methodology often fails to discriminate between allelic variants exhibiting similar amplicon length but distinct genetic sequences (*Halász et al., 2008*; *Gordillo et al., 2015*; *Kodad et al., 2008*; *López et al., 2004*).

To address this limitation, we developed a CAPS molecular marker system designed to identify sequence-level polymorphisms in the S-RNase C2–C3 intragenic region. This region was selected as it includes the RHV sequence; the latter presumed to mediate the recognition of genetically related S-RNase/SFB complexes (*Matton et al., 1997*; *Ushijima et al., 1998*). The RHV region is located at the surface of the S-RNase protein and is thought to play a key role in the recognition of self- and non self-pollen (*Matton et al., 1997*; *Ushijima et al., 1998*); it is therefore considered an adequate proxy of the self-incompatibility response in *P. serotina*. Furthermore, our results indicated that the C2–C3 intragenic region represents the most polymorphic segment of the *P. serotina* S-RNase gene. In fact, Shannon entropy indices and Ka/Ks ratios revealed that the highest degree of amino acid sequence variation in the gene concentrates in the RHV region.

The CAPS marker system designed in this study proved highly effective in the identification of S-RNase alleles. When employed to genotype an initial core-set of 15 *P. serotina* accessions representing a comprehensive sample of the species' S-RNase allelic diversity in the Ecuadorian highlands (*Gordillo et al., 2015*), the system enabled the unequivocal identification of 18 S-RNase alleles. From these, seven alleles ($S_{12}$–$S_{18}$) could not be identified when using the Intron I molecular marker system (*Gordillo et al., 2015*), and four ($S_{15}$–$S_{18}$) could not be detected when amplifying, extracting and sequencing the full-length S-RNase genes for the aforementioned 15 individuals. Furthermore, three additional alleles ($S_{19}$–$S_{21}$) were identified when characterizing a set of 7 individuals used for pollen-tube formation analyses. The effectiveness of our CAPS marker system in the determination of S-RNase diversity can be ascribed to three important factors. In the first place, the use of species-specific primers to amplify the C2–C3 intragenic region allowed for a higher resolution and differentiation of length-based allelic variants. Size differences in the C2–C3 intragenic region are attributable to differences in the lengths of the Intron II and RHV sequences (i.e., the most polymorphic regions of the S-RNase gene) which are easier to resolve via gel electrophoresis as low-molecular weight products. By contrast, the amplification of the full-length S-RNases yield high-molecular weight amplicons of similar size which are not easily isolated. The inability to purify these amplicons independently may lead to the masking-off of allelic differences resulting in allele-misidentification during sequencing (i.e., pooling of two or more allelic variants yields one consensus sequence as seen in this study). In second place, the use of 3 restriction enzymes is a powerful tool to exploit and elucidate the intrinsic, polymorphic nature of the C2–C3 intragenic region. This feature is especially useful when two S-alleles exhibit similar amplicon lengths (i.e., gel electrophoresis does not allow the clear resolution of size-differences of a few base pairs). Accordingly, it offers an opportunity to identify heterozygotes when an individual's S-alleles are similar in sequence length (*Moriya et al., 2007*). Finally, the *in-silico* design of our CAPS molecular system allowed us the possibility to simulate multiple digestions in order to identify the most informative restriction enzymes. The efficiency of our 3 enzyme CAPS marker system is evident when comparing it to CAPS systems developed for haplotyping the S-Locus in other species from the Roseaceae family. For instance, for the identification of 17 S-alleles in European pear cultivars it was necessary to employ

11 different enzymes (*Moriya et al., 2007*), whereas for the identification of 22 S-alleles in Japanese apple cultivars, 17 different enzymes were required (*Kim et al., 2008*).

## S-Locus allelic diversity and its influence over GSI in *P. serotina*

As evidenced in this study, the S-RNase of *P. serotina* shares a high degree of identity with S-RNases from related species for which the S-Locus has been reported to control GSI. This finding would suggest that the S-Locus also plays a crucial role in determining the reproductive patterns of *P. serotina*. To add evidence to this theory, we investigated pollen-tube formation in crosses between compatible and incompatible individuals (as defined by their S-allelic composition via CAPS).

Our results demonstrated that crosses between heterozygote individuals with contrasting S-alleles result in normal pollen tube formation and growth (i.e., pollen tubes extend along the style and reach the ovary). In these crosses, the number of pollen tubes reaching the ovary was abundant. By contrast, in pistils from crosses between heterozygote individuals with exactly similar S-allele identities (including self-pollinations), the vast majority of pollen tubes showed morphological alterations (i.e., swelling of pollen tube tips) and arrested development (Fig. 5). In a number of incompatible crosses, a variable number of pollen tubes could reach the lower-third section of the style, albeit at a comparatively lower density than in compatible crosses. These pollen tubes also showed swollen tips, a morphological alteration that is caracteristic of aberrant pollen tubes (*Oukabli et al., 2000*; *Ludwig et al., 2013*; *Cachi et al., 2014*; *Estaji et al., 2016*; *Radunić et al., 2017*). For two specific incompatible crosses (i.e., 22 × 17 and 17 × 17), however, a substantial number of pollen tubes could reach the ovary. This unexpected result could be evidence of a full or partial breakdown of the *P. serotina* GSI system.

The breakdown of GSI is not an uncommon phenomenon in *Prunus* (*Yamane et al., 2001*; *Hauck et al., 2002*; *Tobutt et al., 2004*; *Zarrinbal et al., 2018*; *Guerra et al., 2020*). In fact, in a previous investigation of GSI in *P. serotina*, *Donovan (1969)* reported that selfings of specific genotypes led to normal fertilization, embryo formation and fruit set. *Donovan (1969)* concluded that the incompatibility reaction in *P. serotina* was not as strong as in other members of the genus, and the author could not rule out self-compatibility in the species. In other *Prunus* species, the partial or full loss of self-incompatibility has been primarily ascribed to the presence of non-functional S-haplotypes caused by mutations in the S-RNase (*Hauck et al., 2002*; *Hauck et al., 2006*; *Li et al., 2020*) or SFB genes (*Wünsch & Hormaza, 2004*; *Tao et al., 2007*; *Wu, Li & Li, 2013*; *Muñoz Espinoza et al., 2017*) that lead to loss of functionality. In this study, we have focused exclusively on investigating the phenotypic behavior of pollen tube formation patterns in compatible and incompatible crosses using S-RNase diversity as proxy for S-Locus identity. However, GSI in *Prunus* is attributed to the interaction and recognition of self/non-self S-RNase/SFB complexes (*Matsumoto & Tao, 2016*), and a clearer understanding of GSI and self-compatibility in *P. serotina* will also require a profound investigation of SFB diversity. Furthermore, a concise understanding of the extent of GSI in *P. serotina*, and the extent to which the S-Locus influences the reproductive patterns of the species, will also need to be explored and validated with fertilization and fruit set experiments *in vivo*.

## CONCLUSIONS

In this study, our objective was to investigate if the *P. serotina* S-Locus controls gemetophytic self-incompatibility (GSI) in the species. To this end, we characterized the molecular structure of the *P. serotina* S-RNase gene and developed a CAPS molecular marker system to unequivocally determine S-RNase alleles. Experimental crosses between cross-compatible and cross-incompatible S-haplotypes (as determined by CAPS) were then used to evaluate pollen tube-formation patterns in-vivo. Our results demonstrate that crosses between heterozygote individuals with contrasting S-haplotypes result in normal pollen tube formation and growth (i.e., pollen tubes extend along the style and reach the ovary). Accordingly, crosses between individuals with the same S-haplotype typically lead to the morphological alteration of pollen tubes and arrested development during fertilization. Notwithstanding, we also report that in some (suspected) incompatible crosses, pollen tubes can reach the ovary. The latter hints at the possibility of genotype-specific breakdown of GSI in the species.

## ACKNOWLEDGEMENTS

The authors would like to acknowledge the technical assistance offered by researchers at the Plant Biotechnology Laboratory (COCIBA, USFQ), as well as Dr. Pieter van 't Hof for his technical assistance in the initial stages of this research project.

### Funding

This research was funded by the International Foundation for Science (IFS) with grant # D/5054-2. The funders had no role in study design, data collection and analysis, decision to publish, or preparation of the manuscript.

### Grant Disclosures

The following grant information was disclosed by the authors:
International Foundation for Science (IFS): # D/5054-2.

### Competing Interests

The authors declare there are no competing interests.

### Author Contributions

- Milton Gordillo-Romero, Lisa Correa-Baus, Verónica Baquero-Méndez and Carlos Vintimilla performed the experiments, analyzed the data, prepared figures and/or tables, and approved the final draft.
- María de Lourdes Torres, Jose Tobar, Andrés F. Torres conceived and designed the experiments, authored or reviewed drafts of the paper, and approved the final draft.

## Data Availability

The DNA sequences corresponding to the *P. serotina* S-alleles are available at GenBank: MN098833, MN098834, MN098835, MN098836, MN098837, MN098838, MN098839, MN098840, MN098841, MN098842, MN098843, MN098844, MN098845 and MN098846.

The raw data corresponding to the pollen tubes development within the style are available as Supplemental File.

## Supplemental Information

Supplemental information for this article can be found online at http://dx.doi.org/10.7717/peerj.9597#supplemental-information.

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
