# Peer review of "Gametophytic self-incompatibility in Andean capuli (Prunus serotina subsp. capuli): allelic diversity at the S-RNase locus influences normal pollen-tube formation during fertilization"

_PeerJ, doi:10.7717/peerj.9597_

## Round 0.1 · original submission · Major Revisions

The reviewers suggest improvements in the reporting of methods (e.g., PCR primer sequences) and the presentation and discussion of results. As well, additional references should be considered.

Reviewer 1 ·

Basic reporting

This is an interesting paper that addresses gametophytic self-incompatibility in tetraploid Prunus serotina. The subject is relevant since no previous molecular studies have been performed in this subject in this species. The article is easy to read and straightforward.

However, there are some issues that should be addressed to merit publication:
- Literature references. Even if self-incompatibility has not been studied at the molecular level in this crop, some key references are missing in the manuscript. One is a PhD. thesis [Self-and cross-incompatibility in black cherry (Prunus serotina)] from 1969 and freely available at https://ufdc.ufl.edu/UF00097757/00001. Some other references related to the species could be added. For example: Pairon et al., J. AMER. SOC. HORT. SCI. 133(3):390–395. 2008 and others. In addition, work performed in tetraploid sour cherry should also be cited (example Tobutt, K.R., Bošković, R., Cerović, R. et al. Identification of incompatibility alleles in the tetraploid species sour cherry. Theor Appl Genet 108, 775–785 (2004)) since this is relevant to the results presented here.

- Introduction.
• Arboreal. I would recommend replacing this word in the document by “woody perennial” or similar.
• Replace “berries” by either “fruits” or “drupes”, which is the correct botanical term for the fruits of Prunus.
• The ploidy of Prunus serotina should be described with a citation in the introduction section. In addition, information describing the situation of Prunus serotina in the genus Prunus should also be added. (see for example Pairon et al., J. AMER. SOC. HORT. SCI. 133(3):390–395. 2008.).

- Materials and methods
• Line 86. It is not clear why the authors for the pollination experiments use unknown trees from a private orchard instead of using the same plant material used for the molecular analyses. This should be explained
• Line 155; change “fertilized pistils” by “pollinated pistils”
• Line 158. This seems inaccurate since the BBCH scale uses numbers not letters. I assume this refers to the balloon stage which could be either stage 59 or 60.

- Results
• Figure 5 is of low quality. I recommend improving the quality of the figure so the pollen tubes can be properly seen.
• Line 267. It is not possible to appreciate swelling of the tip of the pollen tubes in Figure 5a.
• Raw data of pollen tubes. Looking at the excel file with the number of pollen tubes at different levels of the style, pollen tubes in significant numbers are found at the third level of the style in many incompatible crosses. This should be explained with detail since it is not clear from these data that the crosses are indeed incompatible. Crosses in the field of these combinations should be made to confirm this point.

- Discussion
• Why only two alleles per genotypes are obtained if the species is tetraploid? Should not we expect some cases with more than two alleles as observed in tetraploid sour cherry?

Experimental design

The paper describes original primary research that falls within the aims and scope of PeerJ.
The experimental design is appropriate to address the main research question and the methods are described with sufficient detail. However, there is a main problem with the results presented since no crosses in the field have been performed and, consequently, we cannot rule out that self-incompatibility is not complete; in fact, in the raw data we can see in several self-incompatible crosses, pollen tubes reaching the base of the style. I recommend performing those crosses in the field.

Validity of the findings

The paper is of value but, in my opinion, it would be necessary to validate the results obtained with crosses performed in the field.

Additional comments

The paper is of interest but it needs to add additional references and properly address the finding of pollen tubes at the base of the style in putatively incompatible crosses; those should be validated by pollinations made in the field of those same pollinations made in the laboratory.

·

Basic reporting

This study was aimed at identifying S-Locus encoded S-RNase alleles of P. serotina
and to use this data for identifying compatible accession to overcome self-incompatibility barrier in this species. The authors successfully used S-RNase allele sequences for the identification of S-haplotypes in P. serotine. The paper is well written and experiments are well designed and analyzed. It is interesting finding that tetraploid P. serotina shows cross-compatibility patterns and GSI did not breakdown as a result of hetero-allelic pollens.

I recommend the revision of this manuscript. Here are my specific suggestions and comments for improvement of this manuscript:

1. Provide the sequence of the primers used for PCR amplification in the Methods section. Currently, there are two back-references used [PaConsI-F (Sonneveld et al. 2003) and EM-PC5consRD (Sutherland et al. 2004) primers]. The primers should also be mentioned in the legend of the figures/supplementary figure (gels), so that it becomes an independent and easily accessible entity.

2. Are these 15 P. serotina accessions used in this study available in any germplasm bank for other researchers? please give details of their availability and passport data. This information should also be provided /updated with the sequence accession numbers in the NCBI.

3. Some repetition in results and discussion sections can be avoided (for example line 274-285 as previously described in results).

4. Table 4: contains proposed crosses according to the S-allelic composition of the 7 P. serotina trees genotyped with an expected outcome. The authors can add one more column of the results from crossing. It makes sense to compare results with their hypothesis.

5. Figure 4: Gel images need better annotations. The gel lanes should have marked for the restriction enzyme used corresponding to each lane. It is very difficult to go back and forth from Table 1 to figure out what is actually going on in the figures. The S alleles numbers are not organized and are haphazard that adds an extra complication to the figures.

6. Figure 6: the text of sequence alignment should be larger and legible.

7. Supplementary_File_Raw_Data_1: Column labels should be descriptive: what do you mean by First Third/ second Third/ Third Third: is this the # of the cross showing pollen tube growth along the style length. A legend on the Top of the Supplementary Table will be useful.

Are these replicates? are these parallel experiments. What the numerical count means (seeds or pollen tubes?).

8. Supplementary_Uncropped_gels: each of the gel’s pictures needs an independent label and legend and they should be labeled properly.

9. Table S1: DNA sequences for all S alleles are in the NCBI GenBank database (accession numbers: MN098833-MN098846): Checked

Experimental design

-The experiments are well designed and analyzed.

-This manuscript extends the S-haplotype mapping to additional accessions. The concepts and findings are not novel.

-Methods are described in sufficient detail.

-Additional information on the primers and information about the availability of the germplasm from 15 accessions of P. serotina will benefit the research community. Such information and relevant passport data should be updated to sequence submitted to NCBI.

Validity of the findings

-All relevant data is provided and conclusions are well-drawn.

- The figures and supplementary figs need improved labeling and legends for legibility.

Additional comments

The authors successfully used S-RNase allele sequences for the identification of S-haplotypes in P. serotine. The paper is well written and experiments are well designed and analyzed. It is interesting finding that tetraploid P. serotina shows cross-compatibility patterns and GSI did not breakdown as a result of hetero-allelic pollens.

---

## Round 0.2 · accepted · Accept

Both reviewers agree on the scientific soundness and high quality of your revised manuscript.

Reviewer 1 ·

Basic reporting

The authors have successfully addressed my previous comments on the manuscript.

Experimental design

The authors have successfully addressed my previous comments on the manuscript.

Validity of the findings

The authors have successfully addressed my previous comments on the manuscript.

Additional comments

The authors have successfully addressed my previous comments on the manuscript.

·

Basic reporting

Clear and concise.

Experimental design

Methods have been described sufficiently

Validity of the findings

Supported by data and well-articulated.

Additional comments

All the previous suggestions have been addressed.